# Weakening of the cognition and height association from 1957 to 2018: Findings from four British birth cohort studies

David Bann[1]*, Liam Wright[1], Neil M Davies[2,3]†, Vanessa Moulton[1]†

[1]Centre for Longitudinal Studies, Social Research Institute, University College London, London, United Kingdom; [2]MRC IEU, University of Bristol, Bristol, United Kingdom; [3]K.G. Jebsen Center for Genetic Epidemiology, Department of Public Health and Nursing, NTNU, Norwegian University of Science and Technology, Trondheim, Norway

## Abstract

**Background:** Taller individuals have been repeatedly found to have higher scores on cognitive assessments. Recent studies have suggested that this association can be explained by genetic factors, yet this does not preclude the influence of environmental or social factors that may change over time. We thus tested whether the association changed across time using data from four British birth cohorts (born in 1946, 1958, 1970, and 2001).

**Methods:** In each cohort height was measured and cognition via verbal reasoning, vocabulary/comprehension, and mathematical tests; at ages 10/11 and 14/17 years (N=41,418). We examined associations between height and cognition at each age, separately in each cohort, and for each cognitive test administered. Linear and quantile regression models were used.

**Results:** Taller participants had higher mean cognitive assessment scores in childhood and adolescence, yet the associations were weaker in later (1970 and 2001) cohorts. For example, the mean difference in height comparing the highest with lowest verbal cognition scores at 10/11 years was 0.57 SD (95% CI = 0.44–0.70) in the 1946 cohort, yet 0.30 SD (0.23–0.37) in the 2001 cohort. Expressed alternatively, there was a reduction in correlation from 0.17 (0.15–0.20) to 0.08 (0.06–0.10). This pattern of change in the association was observed across all ages and cognition measures used, was robust to adjustment for social class and parental height, and modeling of plausible missing-not-at-random scenarios. Quantile regression analyses suggested that these differences were driven by differences in the lower centiles of height, where environmental influence may be greatest.

**Conclusions:** Associations between height and cognitive assessment scores in childhood-adolescence substantially weakened from 1957–2018. These results support the notion that environmental and social change can markedly weaken associations between cognition and other traits.

**Funding:** DB is supported by the Economic and Social Research Council (grant number ES/M001660/1); DB and LW by the Medical Research Council (MR/V002147/1). The Medical Research Council (MRC) and the University of Bristol support the MRC Integrative Epidemiology Unit [MC_UU_00011/1]. NMD is supported by an Norwegian Research Council Grant number 295989. VM is supported by the CLOSER Innovation Fund WP19 which is funded by the Economic and Social Research Council (award reference: ES/K000357/1) and Economic and Social Research Council (ES/M001660/1). The funders had no role in study design, data collection and analysis, decision to publish, or preparation of the manuscript.

*For correspondence: david.bann@ucl.ac.uk

†Joint Senior Authors

Competing interest: The authors declare that no competing interests exist.

### Editor's evaluation

This paper provides valuable evidence for a weakening of the association between cognitive ability and height from 1957 to 2018 in the UK. The authors find the strength of the association declined over this time frame. These associations were further attenuated after accounting for proxy measures of social class. This paper is a solid contribution to debates about how genetic, environmental, and social factors have affected the joint distribution of height and cognitive ability over the last 60 years.

## Introduction

Cognitive ability is a potentially important determinant of health, with associations between higher cognition and favorable subsequent health outcomes repeatedly documented in childhood, adolescence, and across adulthood (*Deary and Batty, 2007*). This body of evidence—termed by some as 'cognitive epidemiology'—includes associations of lower cognition and anthropometric measures such as shorter height (*Marioni et al., 2014*; *Keller et al., 2013*). Shorter height is in turn associated with a range of disease outcomes and, like cognitive development, is influenced by early life factors (*Monteiro and Victora, 2005*; *Walker et al., 2007*; *Perkins et al., 2016*; *Batty et al., 2009*; *Sudfeld et al., 2015*).

Recent studies have suggested that links between higher cognition and taller height can be large or to a sizable extent explained by shared genetic influences (*Marioni et al., 2014*; *Keller et al., 2013*; *Silventoinen et al., 2012*; *Vuoksimaa et al., 2018*), with others suggesting a role of assortative mating (*Keller et al., 2013*; *Beauchamp et al., 2011*). *Keller et al., 2013* for example estimated that approximately half of the correlation was accounted for by assortative mating, and half by shared genetic factors. However, this evidence is largely derived from the analysis of single cohorts. Historical data can yield insight into the importance of the environment, even for highly heritable traits. Indeed, while height is highly heritable (*Jelenkovic et al., 2016*), substantial increases in average height occurred in the 20th century (*Meredith, 1976*; *Cole, 2000*), likely due to improved nutrition and a declining impact of early life infectious disease (*Walker et al., 2007*; *Perkins et al., 2016*) which in particular affected early life leg growth (*Cole, 2000*). Cognitive test scores are also highly heritable (*Plomin and Deary, 2015*) yet have also increased in the 20th century (the 'Flynn' effect) (*Bratsberg and Rogeberg, 2018*) suggesting that both factors are responsive to environmental change. Shared developmental processes may affect both outcomes (*Sudfeld et al., 2015*; *Mansukoski et al., 2020*),

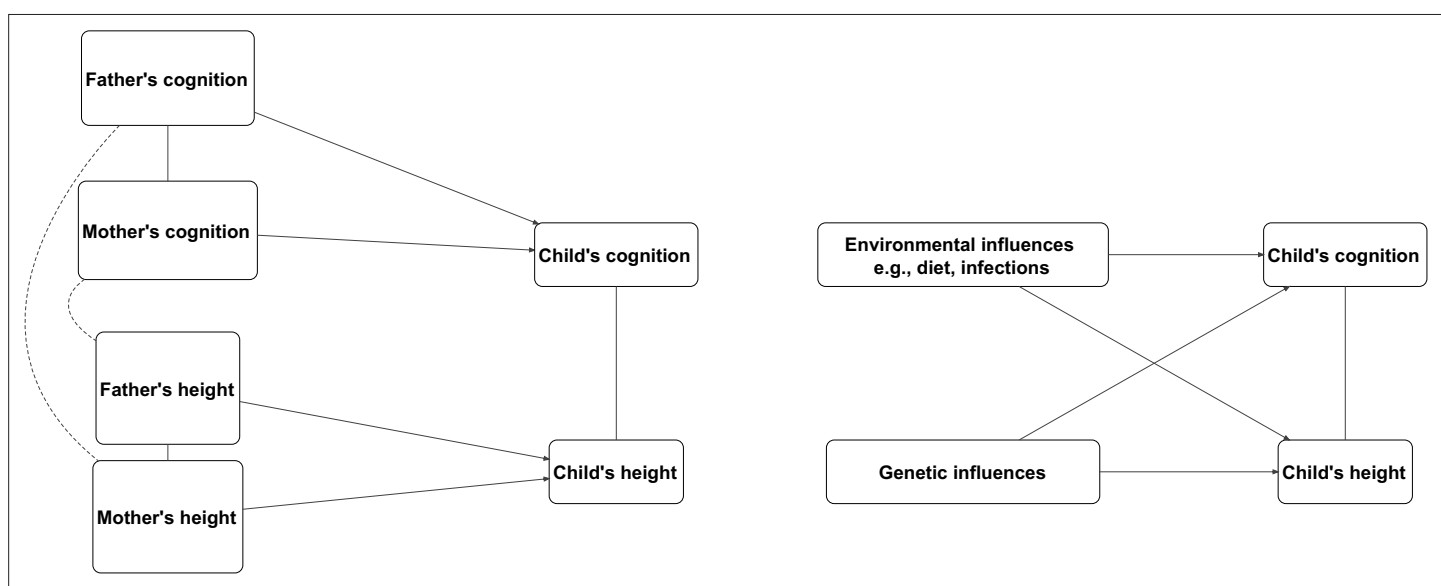

**Figure 1.** Illustrative causal diagrams of two alternative processes which may either (or both in some combination) generate associations between child cognition and height: (right) shared genetic and environmental factors and (left) assortative mating of parental cognition and height (within and across each trait).

**Table 1.** Participant characteristics: Data from four British cohort studies.

| | Cohort study, birth year | | | |
|---|---|---|---|---|
| | 1946 c | 1958 c | 1970 c | 2001 c |
| **Cohort characteristics and outcomes** | | | | |
| Sample size | 5,348 | 17,490 | 17,617 | 17,057 |
| Males (%) | 52.6% | 51.7% | 51.9% | 51.4% |
| Height at 11 years, mean (SD) z-score | −0.41 (1.03) | −0.25 (1.03) | −0.18 (1.01) | 0.29 (1.00) |
| Height at 16 years, mean (SD) z-score | −0.54 (1.03) | −0.35 (1.02) | −0.14 (1.16) | 0.27 (0.98) |
| | | | | |
| **Confounding variables** | | | | |
| Social class | | | | |
| I Professional | 3.0% | 5.4% | 6.1% | 5.0% |
| II Managerial and technical | 14% | 18.2% | 23.6% | 43.1% |
| III Skilled Manual | 46.7% | 43.4% | 44.5% | 21.7% |
| III Skilled Non-Manual | 9.1% | 9.3% | 9.2% | 13.2% |
| IV Partly Skilled | 19.0% | 17.6% | 12.5% | 14.0% |
| V Unskilled | 8.2% | 6.1% | 4.1% | 2.9% |
| | | | | |
| Maternal education, % post compulsory | 20.86% | 25.1% | 34.5% | 52.8% |
| Maternal height, mean (SD) cm | 160.81 (6.38) | 161.99 (6.45) | 161.28 (6.63) | 164.14 (6.93) |
| Paternal height, mean (SD) cm | 172.68 (8.07) | 174.47 (7.42) | 175.21 (7.51) | 178.28 (7.23) |

Note: Data are imputed (32 imputations); see methods for details and exact age of measurement in each cohort.

such as growth influencing height and brain size (*Vuoksimaa et al., 2018*). Improvements in nutrition and other shared developmental determinants may have weakened associations between cognition and height (*Gale, 2005*), as suggested by a weakening of links between low birth weight and cognition from 1958–2001 (*Goisis et al., 2017a*). A declining association between cognition and height has been suggested in the analysis of a subsample of eastern Danish male conscripts (1939–1967) (*Teasdale et al., 1989*), yet data from more heterogeneous and contemporary samples is required to strengthen conclusions and aid generalizability.

We conducted a cross-cohort study to investigate if the associations of cognition with child-adolescent height have systematically changed over time (1957–2018). We used four British birth cohort studies, each containing prospectively ascertained assessments of cognition and height. We hypothesized that improvements in the shared determinants of both cognition and height would lead to the weakening of associations across time; see *Figure 1* for a causal diagram. We also hypothesized that cognition-height associations would be strongest amongst those in the lowest height centiles, where environmental influences on impaired height development may be particularly pronounced.

## Methods
### Study samples
Four birth cohort studies conducted in Britain with participants followed-up in childhood and adolescence were used. These were born in 1946 (MRC National Survey of Health and Development, 1946 c), 1958 (National Child Development Study, 1958 c), 1970 (British Cohort Study, 1970 c), and 2000/02 (Millennium Cohort Study, 2001 c); further details are provided in the cohort profiles (*Wadsworth et al., 2006*; *Power and Elliott, 2006*; *Elliott and Shepherd, 2006*; *Connelly and Platt, 2014*). To aid comparability, analyses were restricted to singleton births born in England, Scotland, and Wales.

Sample sizes for analyses, in each cohort and age, are shown in *Table 1*. Each study and sample sweep used in this manuscript has received ethical approval and obtained parental/participant consent according to guidance that was in place at the time of data collection from 1957–2018. These historic datasets are made available for anonymized observational research which can be published on a range of topics by bona fide researchers; further details are provided elsewhere (*Shepherd and Gilbert, 2019a*; *Centre for Longitudinal Studies, 2014*; *Shepherd and Gilbert, 2019b*) ethical review boards are as follows: 1946 c (Queen Square Research Ethics Committee: 14/LO/1073; and the Scotland A Research Ethics Committee: 14/SS/1009), 1958 c (London – Central: 12/LO/2010), 1970 c (London-Central Research Ethics Committee: 14/LO/0371), 2001 c (London – Central MREC: 13/LO/1786).

## Measures
### Height
As described elsewhere (*Johnson et al., 2015*), height was obtained at the following ages in each study: ages 11 and 16 in the 1946 c and 1958 c; ages 10 and 16 in 1970 c; ages 7, 11, 14, and 17 in 2001 c. Height was directly measured in all instances, except age 16 years in 1970 c in which either self-report or objectively measured height was obtained (see sensitivity analyses below). To facilitate cross-cohort comparisons in our main analysis height variables were converted to age and sex-adjusted z-scores using growth reference charts (the UK90 reference panel; the *childsds* package in R *Vogel, 2018*). As a robustness check, we also converted height to cohort-specific z-scores at each age (i.e. without reference to an external panel such as the UK90 as used in the main analysis), percentile ranks, and ridit scores (see below).

### Cognitive function
Given differences in the cognitive tests administered to each cohort (*Moulton et al., 2020*), we screened and identified measures that captured the same broad cognitive domains in childhood and adolescence. Assuming the relative ranking was similar, this would enable valid cross-cohort comparisons of associations with outcomes. Differences in the cognitive tests completed could potentially bias cross-cohort comparisons of cognition-height associations. Thus, we used multiple tests assessing different cognitive domains, measured at two different ages in order to examine whether our main findings were robust to the cognitive domain tested or the age chosen. A summary is provided below—information on the tests used, procedures, and scoring is provided in *Supplementary file 1*; further information on the origins and technical details of these tests is available from elsewhere (*Moulton et al., 2020*).

*Age 10/11* years: verbal reasoning test scores. These capture general verbal ability, verbal knowledge, and reasoning. In the 1946 c and 1958 c, this was measured using the verbal items from the General Ability Test (GAT) (*Pigeon, 1964*). In the younger cohorts tests from the British Ability Scales (BAS), the Word Similarities test—the precursor to the Verbal Similarities (BAS II)—were administered in the 1970 c and 2001 c, respectively (*Elliot et al., 1978*; *Elliott et al., 1996*).

*Age 14/16* years: reading/vocabulary test scores. The Watts-Vernon Reading Test (WVRT) was administered in the 1946 c and the Reading Comprehension Test in the 1958 c was devised by the National Foundation for Education Research (NFER) to parallel the aforementioned WVRT; both tests measure reading and word comprehension (*Shepherd, 2012*). The Applied Psychology Unit (APU) Vocabulary Test was administered in both the 1970 c and 2001 c and measures word knowledge and meaning (*Closs, 1976a*).

Furthermore, mathematic tests were administered in each cohort and used as an alternative indicator of cognitive performance. These were measured at 15/17 years in each cohort, at 7 years in 2001 c and 10/11 years in the 1946 c, 1958 c, and 1970 c. Age 10/11 years and age 7 years in 2001 c: in the earlier cohorts the mathematics tests were specifically designed for each of the cohorts and measured contemporary number skills and mathematical calculations (1946 c *Pigeon, 1964*, 1958 c *Shepherd, 2012*, 1970 c *Parsons, 2014*, 2001 c *Chaplin Gray et al., 2010*). For the 2001 c an adaptation of the NFER Progress in Maths Test was administered. The Age 15/17 years: mathematical tests devised by the NFER were administered in the 1946 c (*Pigeon, 1964*) and 1958 c (*Shepherd, 2012*) and the APU Arithmetic Test in the 1970 c (*Closs and Hutchings, 1976b*); all tested numerical skills and problem-solving, along with geometry and trigonometry in the earlier cohorts. Number

Analogies, a short version of the Quantitative Reasoning Battery (GL Assessments) was administered in the 2001 c.

Since test performance differs by age (*Moulton et al., 2020*), we constructed age-adjusted cognition measures by exporting the residuals from a linear regression model with test score as the outcome and age as the exposure variable (linear term). At age 16 years in 1970 c, vocabulary tests were administered either in the home or in school—since test scores differed by mode, we also adjusted for a binary indicator of mode of administration in these models. Finally, to aid the interpretation of effect sizes—and comparisons across age and cohort—in main analyses, we standardized all cognition measures to ridit scores (*Mackenbach and Kunst, 1997*) individuals in each category were assigned a value based on the proportion of the population with lower cognitive scores. This results in values between 0 and 1 with higher scores equating to better cognitive performance. In regression models coefficients thus show the mean difference in height comparing those with the lowest and highest cognition scores (termed the slope index of inequality in the health inequality literature *Mackenbach and Kunst, 1997*). As a robustness check, we also repeated analyses converting cognition scores to (test and cohort-specific) z-scores and percentile ranks.

## Potential confounders

We hypothesized two sets of confounding variables—family socioeconomic characteristics, and parental height. Childhood socioeconomic position (SEP) was indicated by the father's occupational social class, measured at 10/11 years (highest parental occupational was used for the 2001 c). To aid cross-cohort comparability, the Registrar General's Social Class was used to classify social class—from I (professional), II (managerial and technical), IIIN (skilled nonmanual), IIIM (skilled manual), IV (partly-skilled), and V (unskilled) occupations. Maternal education was also used (ascertained at age six in 1946 c NSHD, at birth in the 1958 c NCDS and 1970c BCS, and at nine months – 14 years in 2001 c MCS) with a binary indicator of whether the mother had left education at the mandatory leaving age (14 years old from 1918, 15 from 1944, and 16 from 1972).

Parental height was reported in early life— six years (1946 c), 11 years (1958 c), 0 & 10 years (1970 c), nine months – seven years (2001 c)—with those with particularly low height removed (height <1.4 m; n=14, 18, 12, 10, respectively in each cohort). Parental cognition was not measured in each cohort, precluding its investigation.

## Statistical analysis

First, we conducted descriptive analyses and estimated bivariate correlations between cognition, social class, and height. Second, we regressed height on individual cognition variables (measured at the same age) adjusting for sex using linear regression since findings were similar in either sex. We did not anticipate childhood cognition and height to be directly causally related, but rather associated due to the influence of common environmental and genetic causes (see *Figure 1*). To test our hypotheses, we sequentially adjusted for parental SEP (class and education) and parental height (mothers and fathers).

We next used quantile regression models (*Koenker and Bassett, 1978*) to investigate whether mean differences observed in linear regression models arose due to differences in the lower quantiles (reflecting impaired height growth, which may be more pronounced in older cohorts). We estimated the cognition-height association at each decile, again adjusting for sex and sequentially for parental SEP and parental height.

To address missing data, multiple imputations by chained equations was completed (separately in each cohort/sex), with 32 imputations performed (burn-in = 10 iterations). To improve the plausibility of the missing at random assumption, additional auxiliary variables were included in imputation models (e.g. childhood BMI and parental education). OLS results were pooled using Rubin's rules (*Rubin, 1976*), while quantile regressions were pooled using the MI-then-bootstrap procedure (*Bartlett and Hughes, 2020*) with confidence intervals derived using the percentile method (200 bootstraps per imputation). The main findings did not differ when complete case analyses were conducted (results available upon request). Since the 1946 c and 2001 c are stratified study designs, analyses using these cohorts were conducted using study-specific design weights.

## Additional and sensitivity analyses

As differences in missing data may bias cross-cohort patterns of association (*Bann et al., 2021*), in addition to using multiple imputations we carried out sensitivity analyses using pattern mixture

modeling *Leurent et al., 2018*; our main OLS models were re-run altering imputed height values by a range of constant values (–1 to + 1 SD). This enabled investigation of the robustness of the main results to non-random missingness in the height data.

We also carried out a series of other robustness checks. First, we repeated all models using different procedures for harmonizing height (raw, z-score, ridit, and rank score) and cognition variables (z-score and rank score). Second, we repeated analyses for males and females separately. Third, as the ethnic mix of the 2001 c greatly exceeds that of older cohorts, we investigated if ethnic composition differences may have contributed to the main inferences drawn by conducting additional analyses restricted to the majority (European ancestry) population. Fourth, as the height at age 16 was collected by self-report for some participants in the 1970 c, we repeated main analyses using only the subset of those with measured height.

The associations of cognition and height could be biased by differences in the measurement of cognition in each cohort. Consequently, we carried out further sensitivity analyses to examine the role of measurement in explaining our results. First, we constrained each set of comparisons to have the same number of cognition response items; those from the cohort with the smallest observed range of scores (e.g. 22 items for verbal scores at 10/11 years; 1970 c had 22 test items, all other cohorts had 40–41 items and were thus rounded down; non-integer values were rounded up or down to the

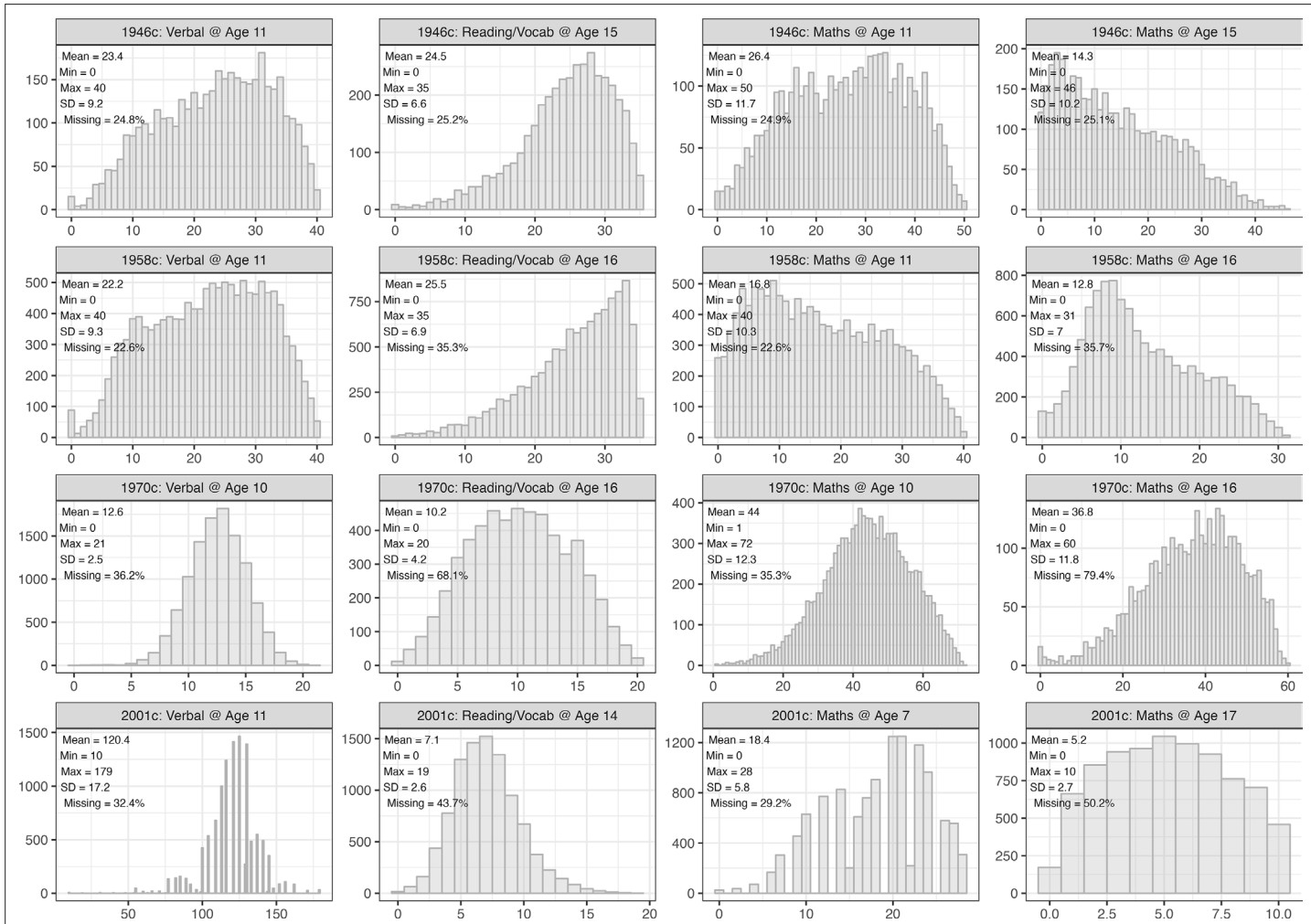

**Figure 2.** Distribution of cognition scores, by cohort and test. Observed data.

The online version of this article includes the following figure supplement(s) for figure 2:

**Figure supplement 1.** Spearman correlation between study variables by cohort.

**Figure supplement 2.** Spearman correlation between paternal and maternal height and education (years) by cohort.

nearest integer; see *Supplementary file 1*). Second, we took random subsets of individual items to produce tests of the same size. This was only possible for the 1970 c and 2001 c, for which item-level data were available.

## Results
### Descriptive statistics

Childhood and adolescent height were greater in each successive cohort (*Table 1*, *Supplementary file 1*). While differences in the cognitive tests administered preclude a comparison of absolute levels of cognition by cohort, the different tests used provided variation within each cohort and at each age (*Figure 2*). Cognitive test scores were strongly moderately positively correlated with each other, with

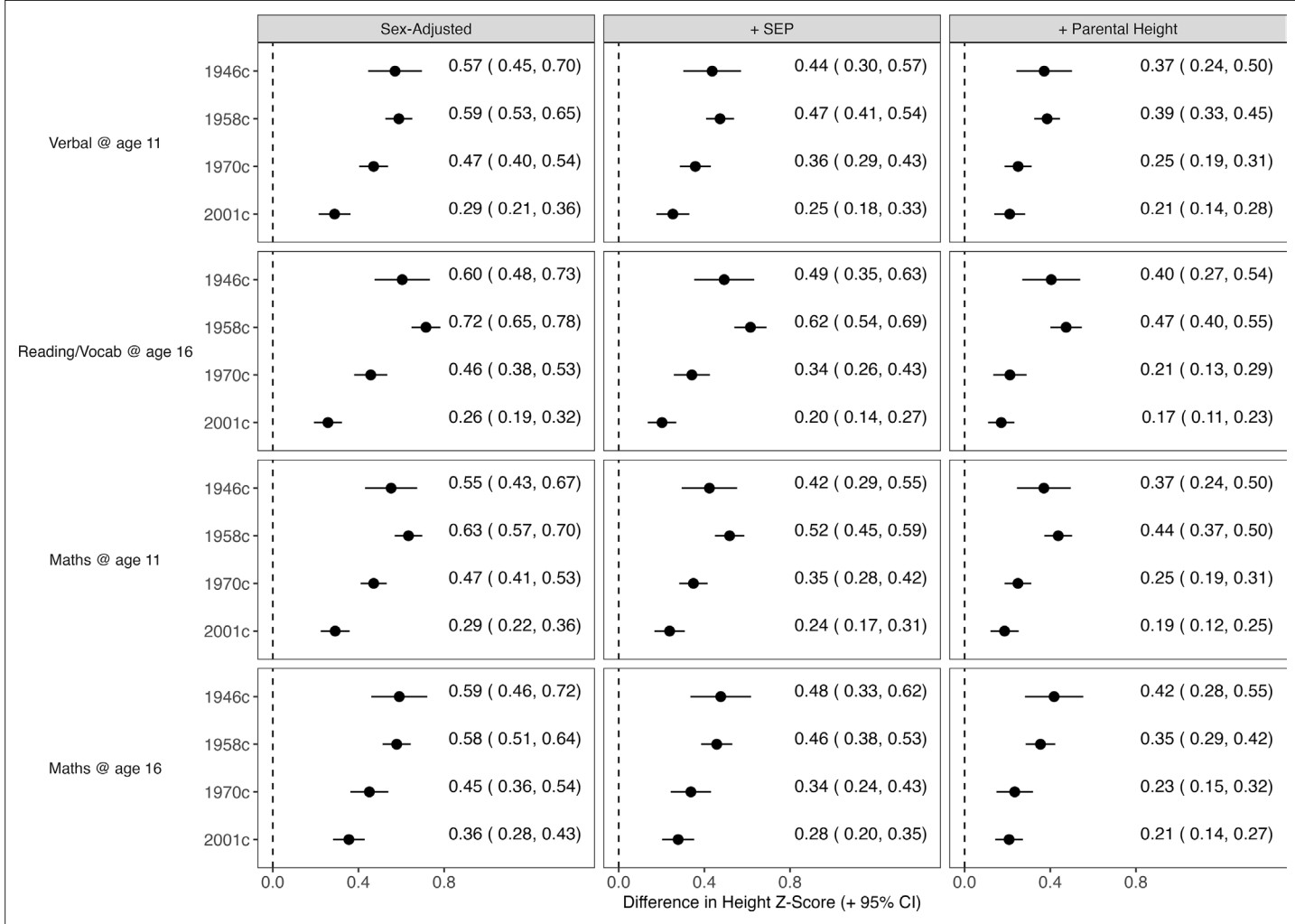

**Figure 3.** Association between height and cognition in four birth cohort studies. Cognition scores are the independent variables and height (age-adjusted z-scores) are the dependent variables; estimates are from separate linear regression models (95% CIs): each estimate shows the mean difference in height (z-score) comparing the highest with lowest cognition test score. Models were sequentially adjusted for sex (left panel); sex, mother's education, and father's social class (middle panel), and additionally for maternal and paternal height (right panel).

The online version of this article includes the following figure supplement(s) for figure 3:

**Figure supplement 1.** Difference in the association (+95% CI) between height and cognition by cohort and test.

**Figure supplement 2.** Attenuation (absolute and proportional) in the association (+95% CI) between height and cognition by cohort, test, and covariates added to models.

**Figure supplement 3.** Association between height and cognition by cohort and test and method used to score cognition measures.

**Figure supplement 4.** Association between height and cognition by cohort and test and method used to score cognition measures.

the size of the correlation weakening across time (*Figure 2—figure supplement 1*). Higher social class was associated with taller height and higher cognition scores; these correlations also appeared to weaken over time, as did positive correlations between maternal and paternal height (*Figure 2—figure supplement 1*). The correlation between parental height and parental education was weakest in the 2001 c, but generally similar in the other cohorts (*Figure 2—figure supplement 2*).

## Main analyses

Higher cognition scores were associated with taller height—this association was found across all cohorts at both 11 and 16 years (*Figure 3*). These associations were stable or slightly larger between the 1946 c and 1958 c cohorts, and weakened thereafter, being substantially weakest in the youngest (2001 c) cohort. For example, after adjustment for sex, the mean difference in height associated with verbal reasoning score at 10/11 years was 0.57 SD (95% CI=0.45, 0.7) in the 1946 c, 0.59 SD (95% CI=0.53, 0.65) in the 1958 c, 0.47 SD (95% CI=0.40, 0.54) in the 1970 c, and 0.29 SD (95% CI=0.21, 0.36) in the 2001 c. Note that cognition scores were rescaled to ridit scores (0–1) to aid comparability: these effect estimates, therefore, show the difference in height comparing the highest with the lowest cognition score. Expressed alternatively, there was a reduction in the correlation coefficient between verbal reasoning scores and height at 10/11 years from 0.17 in 1958 c (0.15–0.20) to 0.08 in 2001 c (0.06–0.10). Testing via the inclusion of cohort*cognition interaction terms also supported this finding (*Figure 3—figure supplement 1*). In particular, estimates from the 1946 c were least precise owing to its smaller sample size, and confidence intervals overlapped with the larger 1958 c. The pattern of weakening association across time was found at both 11 and 16 years and across all measures of cognition (verbal reasoning, maths, and reading/vocabulary).

Associations between cognition and height were partly attenuated after adjustment for parental SEP and partly attenuated further by parental height (*Figure 3*) middle and right panels, respectively. However, the associations did not attenuate completely to zero after this adjustment, and the weakening of the cognition-height association observed across each cohort was similar before and after the adjustment. Attenuation (in absolute terms) when adjusting for parental SEP and parental height was broadly similar in the 1946 c, 1958 c, and 1970 c, but was smallest in the 2001 c (*Figure 3—figure supplement 2*).

## Additional analyses: Quantile regression results

In quantile regression models adjusting for sex, there was evidence that associations between height and cognition were stronger at lower centiles of height, particularly in older cohorts (*Figure 4*). That is, the average (mean) differences reported previously using linear regression models may have been driven by differences in the lower part of the height distribution, as hypothesized. This was especially pronounced in the 1958 c: for verbal scores at age 11, cognitive ability was associated with 0.73 SD (95% CI=0.63, 0.83) greater height at the 10th percentile and 0.47 SD (95% CI=0.37, 0.58) greater height at the 90th percentile. Differences across centiles were less marked in the 2001 c; corresponding figures in 2001c were 0.34 SD (95% CI=0.22, 0.45) and 0.27 SD (95% CI=0.14, 0.41), respectively. Qualitatively similar results were obtained when adjusted sequentially for childhood SEP and parental height.

## Sensitivity analyses

Among participants whose height was measured on at least one occasion, there were small differences in height according to whether an individual was lost to follow-up or not (<0.3 SD). However, repeating main analyses altering imputed values by a constant factor to attempt to account for non-random missingness (i.e. pattern mixture modeling) generally yielded qualitatively similar results, even where the factor was large (i.e.>0.8 SD; *Figure 5*): smallest associations were estimated in the 2001 c, and largest associations were estimated in the 1946 c or 1958 c.

Repeating analyses using different procedures to score cognition (z-score, ridit scores, and percentile ranks) and height (raw, percentile rank, z-scores, ridit scores) variables in order to compare estimates across cohorts also yielded qualitatively similar results to the main OLS analysis, as did limiting analyses with the 2001 c to white participants only, or the 1970 c to those with measured rather than self-reported height at 16 years (results available upon request). Finally, results were similar when

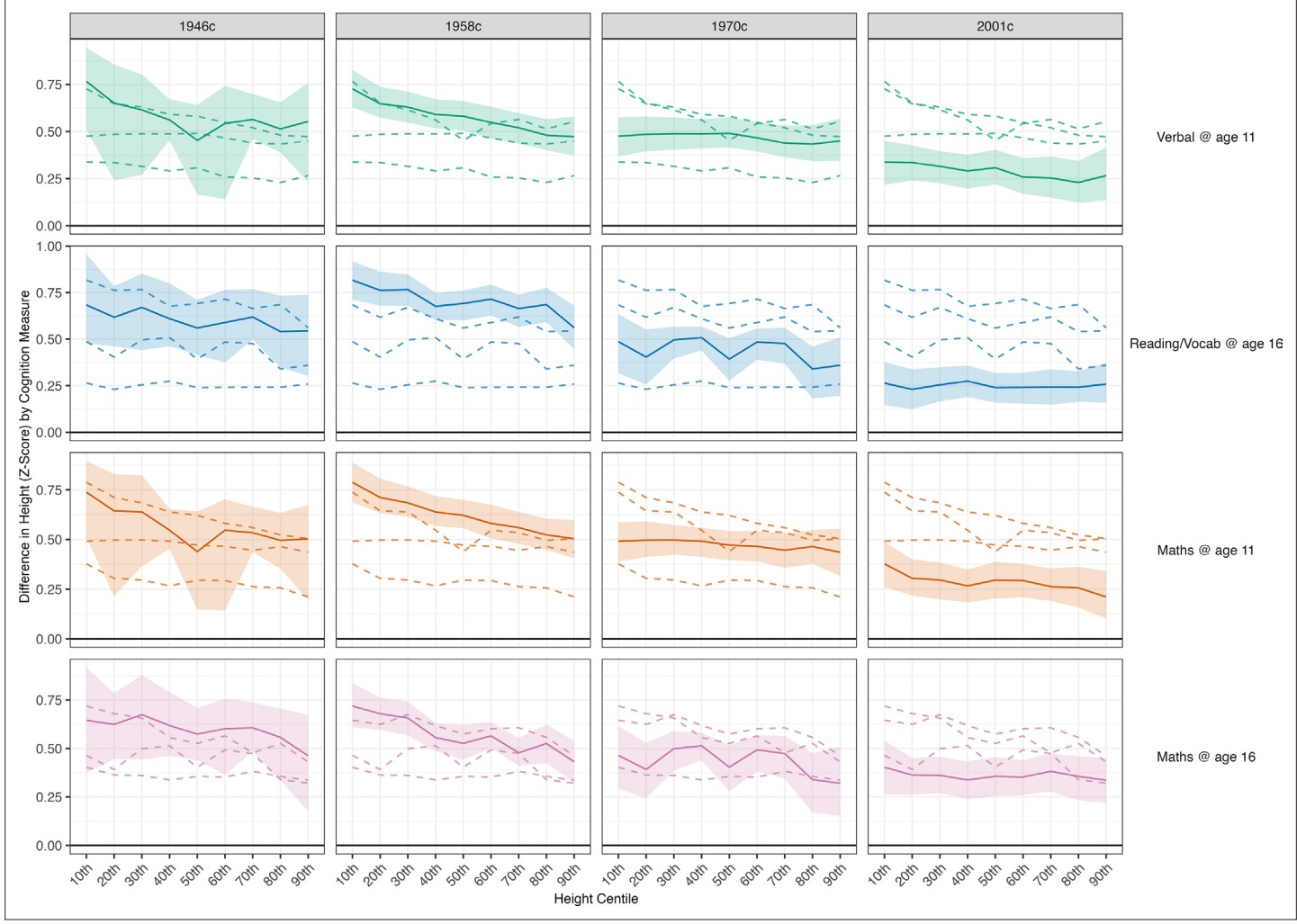

**Figure 4.** Association between height and cognition in four birth cohort studies. Cognition scores are the independent variables and height (age-adjusted z-scores) are the dependent variables; estimates are from separate quantile regression models (95% CIs): each estimate shows the difference in height (z-score) comparing the highest with lowest cognition test score, repeated for each decile of height (x-axis), cohort (panel columns) and measure of cognition (panel rows). Quantile regression results are interpreted similarly to those from linear regression. For example, the 50th centile shows the estimated difference in height at the median (linear regression would show the mean difference). Results were drawn from pooled quantile regression models (32 imputed datasets) with adjustment for sex. Shaded areas are 95% CIs (200 bootstraps per imputed dataset; centile method) for the specific cohort and cognition measure examined. Dashed lines show estimates for other cohorts to aid cross-cohort comparisons.

constraining similar tests to have the same range (*Figure 3—figure supplement 3*) or sampling an equal number of items from similar tests (*Figure 3—figure supplement 4*).

## Discussion
### Summary of findings

The association between higher cognition and height weakened substantially from 1957–2018. Associations were at least half as strong in the youngest (2001 c) compared with the oldest (1946 c and 1958 c) cohorts. Quantile regression analyses suggested that height-cognition associations (and potentially their weakening across time) were driven by differences in lower quantiles of height. Associations were only partly attenuated when controlling for parental SEP or parental height.

Our findings build on earlier work, such as a study of Danish male conscripts (*Teasdale et al., 1989*), which also reported a weakening of the cognition-height association from 1939–1967. We extend these findings by use of more recent data in more generalizable samples, utilizing multiple

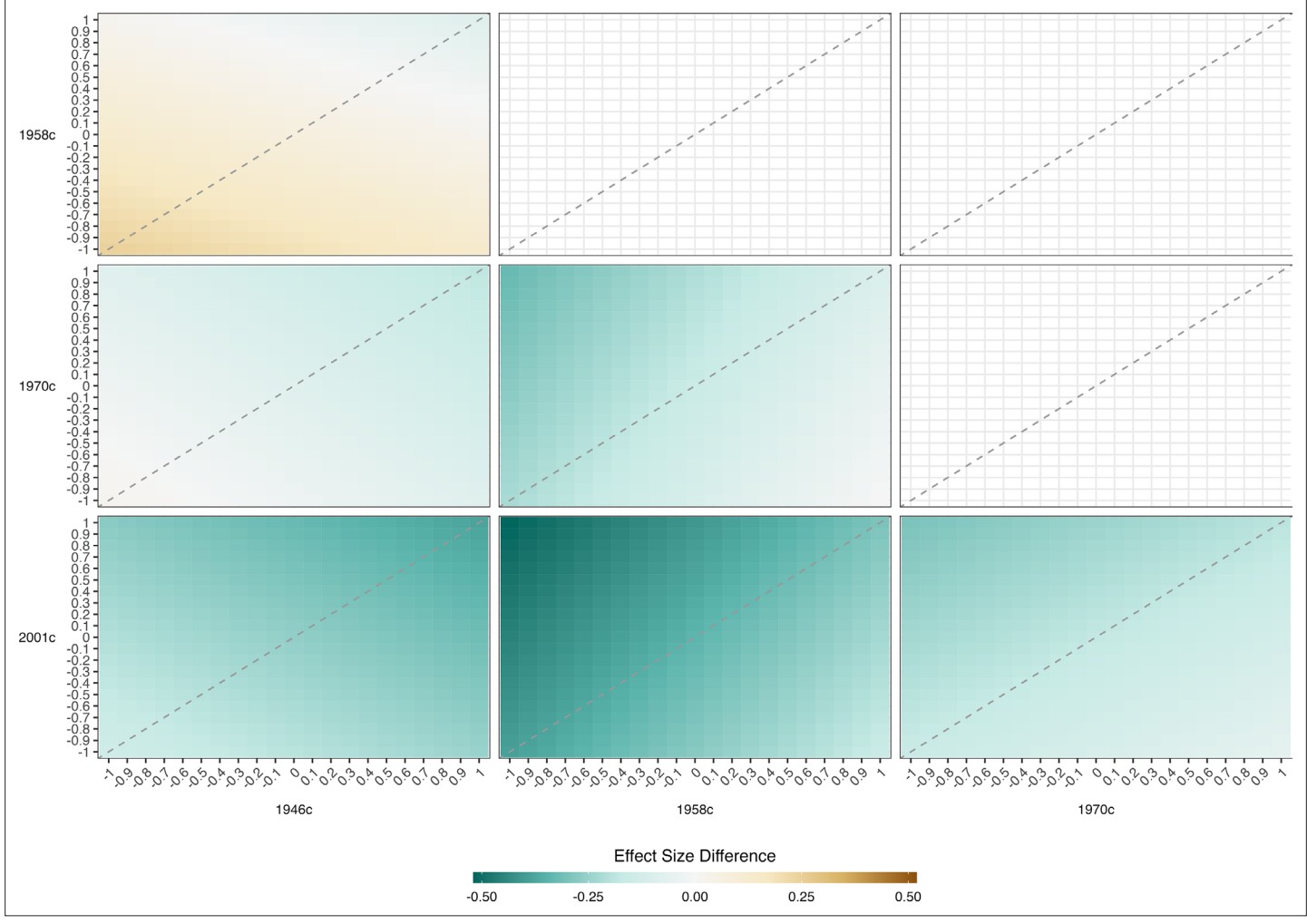

**Figure 5.** Result of pattern mixture models. Difference in the association between height and maths @ age 11 across the cohort. Derived from pooled OLS models (32 imputed datasets) including adjustment for sex. OLS models repeated with imputed height values adjusted by a constant factor given in columns and rows. The color of each square indicates the difference in the association between height and cognition between two cohorts with the coefficient for the cohort on the horizontal subtracted from that on the horizontal (with given adjustment factors used). Height harmonized across cohorts using growth charts from a 1990 UK growth study. Cognition scores harmonized using ridit scoring (range 0–1).

The online version of this article includes the following figure supplement(s) for figure 5:

**Figure supplement 1.** Result of pattern mixture models.

**Figure supplement 2.** Result of pattern mixture models.

**Figure supplement 3.** Result of pattern mixture models.

cognitive tests, undertaking both linear and quantile regression analyses, and accounting for both parental social class and height.

## Explanation of findings

Just as multiple processes could explain why cognition itself has increased across time (the Flynn effect) (*Pietschnig and Voracek, 2015*), multiple processes could explain why associations between cognition and height have weakened across time. First, changes could have been driven by changes across time in the shared determinants of height and cognition. In particular, improvements in nutrition and reductions in the incidence (and/or impact of) infectious disease may have weakened this association, particularly given the stronger magnitude of the association at lower height centiles. This finding is consistent with declines in the socioeconomic patterning of height observed across this time period, as found in this study and reported elsewhere (*Bann et al., 2018*).

We adjusted for multiple indicators of parental SEP available in each cohort (social class and education), and this may only partly capture the socioeconomically-graded environmental factors responsible for this association; further, we cannot fully rule out the potential for residual confounding by SEP itself (i.e. that height and cognition are correlated because of common effects of the familial environment). In particular, an apparent weakening of the correlation between parental social class and offspring cognition across time (*Paterson, 2021*) could have led to more residual confounding and thus stronger associations between height and cognition in older cohorts. In support of this, we found some evidence that attenuation upon adjustment for SEP was more pronounced in older cohorts. However, differences in the measurement of SEP across time may have operated in the other direction. While the SEP measures were constructed to be comparable in each cohort, the social class schema adopted in our study originated in the study of 20th century job roles (*Bland, 1979*); analogously, parental education may have been a more distinguishing trait in earlier cohorts (before the broader expansion of education). Thus, measurement error of SEP and resulting residual confounding may have instead been greater in the younger cohorts. The robustness of the cross-cohort weakening in association after adjustment for parental height (an indicator of genetic liability to taller height) is similarly consistent with the hypothesis that environmental factors which influence early life height and cognition growth may be at least partly responsible for the weakening in association observed rather than genetic factors related to height.

Changes in mating patterns across time could also theoretically contribute to our finding of weakening in the association between cognition and height. A weakening of cross-trait assortative mating by cognition and height across cohorts could drive a weakening in the associations between these factors that we observed. However, we are not aware of empirical evidence for this and are unable to directly test this owing to a lack of data for parental cognition. While we were unable to investigate changes in cognition-height assortative mating, parental height correlations were broadly similar in the 1946 c, 1958 c, and 1970 c and slightly weaker in the youngest cohort; correlations between paternal height and maternal education were also weaker in the youngest cohort. We also lack data to test whether other changes to mating patterns—such as reductions in genetic markers of inbreeding across time (*Nalls et al., 2009*) may explain our findings. In the social science literature, evidence for changes in assortative mating by socioeconomic factors paints a complex picture: studies have suggested either stability (*Henz and Mills, 2018*), increases in assortative mating, or mixed findings (*Schwartz, 2013*). Further research utilizing genetic and parental data is warranted to investigate these possibilities.

The multi-purpose and multidisciplinary cohorts used cognition tests which differed slightly in each cohort. It is, therefore, possible that differences in testing could have either: (1) generated the pattern of results we observed, such that if identical tests were used the association between cognition and height would otherwise have been similar in each cohort; in contrast to previous findings which reported using identical tests *Teasdale et al., 1989*; or (2) biased our results, such that if identical tests were used the decline in the association between cognition and height would have been less marked than we reported. While we cannot directly falsify this alternative hypothesis given our reliance on historical data sources, a number of lines of reasoning suggest that the first scenario is unlikely. First, our results were similar when using four different cognitive tests (spanning mathematical and verbal reasoning); any bias which generated the results we observed should be similarly present across all 4 tests. Other things being equal, one would expect that more discriminatory tests (i.e. those with a greater number of items) would have higher accuracy and thus better measure cognition. Our results were similar when the youngest cohort had similar numbers of unique scores in cognitive tests compared with the oldest cohort (Verbal @ 11 years: n=41 in 1946 c, n=40 in 2001 c) and fewer unique scores (Maths @ 7/11: n=51 in 1946 c, n=21 in 2001 c). Our results were also similar in sensitivity analyses in which the number of response items was set to be the same in each cohort. Higher random measurement error in the independent variable (cognition) would lead to weakened observed associations with the outcome (height) (*Hutcheon et al., 2010*), yet we do not a-priori anticipate that this such error was higher in younger cohorts across all tests in such a manner that would have led to the correlation we observed.

Ensuring comparability of exposure is a major challenge across such large timespans. Reassuringly, our results are consistent with those from a previous study that reported the same tests being used (from 1939 to 1967) (*Teasdale et al., 1989*). However, even seemingly identical tests require

modification across time (e.g. for verbal reasoning/vocabulary there is typically a need to adapt question items due to societal and cultural changes over time in vocabulary and numerical use); further, changes to education such as increases in testing may have led to increasing preparedness and familiarity with testing than in the past even where identical tests are used.

Interestingly, we observed a marked reduction in the correlation between cognitive tests across time (e.g. between verbal and maths scores). This trend has been reported in previous studies (*Sundet et al., 2004*; *Kane, 2000*) and warrants future investigation; it is consistent with evidence that IQ gains across time seemingly differ by cognitive domain (*Pietschnig and Voracek, 2015*) potentially capturing differences across time in cognitive skill use and development in the population. Previous studies using three (1958–2001 c) of the included cohorts have also reported changing associations between cognition (verbal test scores at 10/11 years) and other traits: a declining negative association with birth weight (*Goisis et al., 2017a*) and a change in direction of association with maternal age (from negative to positive) *Goisis et al., 2017b*; each finding has plausible explanations based on changes across time in relevant societal phenomena (improved medical conditions *Goisis et al., 2017a* and changes in parental characteristics *Goisis et al., 2017b* respectfully), yet also cannot conclusively falsify the notion that differences in tests used influences the results obtained. In this paper, we used multiple tests and sensitivity analyses to attempt to address this.

## Strengths and limitations

Strengths of this study include the use of multiple nationally representative longitudinal cohorts from the UK—each with data on multiple cognitive test scores and measured height. We used four cohorts—the latest measure of the youngest cohort was 17 years—since height gains may occur up to 19–20 years (*Rodriguez-Martinez et al., 2020*), further work may be warranted in the future to test if the patterns of association remain in later adulthood. Our analytical strategy is also a strength, enabling us to test the robustness of our findings across multiple ages and multiple domains of cognitive test performance and statistical methods.

Limitations include the potential for bias due to missing data, which is more pronounced in the youngest cohort. However, multiple imputation models were used, with multiple auxiliary variables included to increase the credibility of the 'missing at random' assumption such models entail; and findings from pattern mixture modeling suggested that our main findings were robust to differential missingness of outcome data.

## Conclusions

Associations between higher cognition and taller height in childhood-adolescence weakened markedly from 1957–2018. These findings support the notion that changes to environmental and societal factors across time may substantially weaken associations between cognition and other traits.

## Additional information

### Funding

| Funder | Grant reference number | Author |
|---|---|---|
| Medical Research Council | MR/V002147/1 | David Bann |
| Economic and Social Research Council | ES/M001660/1 | David Bann Vanessa Moulton |
| Economic and Social Research Council | ES/K000357/1 | Vanessa Moulton |
| Medical Research Council | MC_UU_12013/1 | Neil M Davies |
| Medical Research Council | MC_UU_12013/9MC_ UU_00011/1 | Neil M Davies |

The funders had no role in study design, data collection and interpretation, or the decision to submit the work for publication.

## Author contributions
David Bann, Conceptualization, Data curation, Supervision, Funding acquisition, Validation, Investigation, Methodology, Writing – original draft, Writing – review and editing; Liam Wright, Conceptualization, Resources, Data curation, Software, Formal analysis, Investigation, Visualization, Methodology, Writing – review and editing; Neil M Davies, Conceptualization, Investigation, Methodology, Writing – review and editing; Vanessa Moulton, Conceptualization, Data curation, Investigation, Methodology, Writing – review and editing

## Author ORCIDs
David Bann ⬤ http://orcid.org/0000-0002-6454-626X
Neil M Davies ⬤ http://orcid.org/0000-0002-2460-0508
Vanessa Moulton ⬤ http://orcid.org/0000-0002-7400-2522

## Ethics
Each study and sample sweep used in this manuscript has received ethical approval and obtained parental/participant consent according to guidance which was in place at the time of data collections from 1957 to 2018. These historic datasets are made available for anonymized observational research which can be published in a range of topics by bona fide researchers; further details are provided elsewhere; 25-27 ethical review boards are as follows: 1946c (Queen Square Research Ethics Committee: 14/LO/1073; and the Scotland A Research Ethics Committee: 14/SS/1009), 1958c (London - Central: 12/LO/2010), 1970c (London-Central Research Ethics Committee: 14/LO/0371), 2001c (London - Central MREC: 13/LO/1786).

## Decision letter and Author response
Decision letter https://doi.org/10.7554/eLife.81099.sa1
Author response https://doi.org/10.7554/eLife.81099.sa2

# Additional files

## Supplementary files
• MDAR checklist

• Supplementary file 1. Supplementary file for: Weakening of the cognition and height association from 1957 to 2018: findings from four British birth cohort studies. (a) Cognitive measures in four British birth cohort studies used in subsequent cross-cohort analysis: procedures and scoring, with links provided to further information. (b) The numbers of unique scores in each cognition test ) The numbers of unique scores in each cognition test. (c) Descriptive statistics, observed and imputed data. Imputed data drawn from 32 imputations.

## Data availability
Data files are available through the UK Data Service (1958c, 1970c, and 2001c; https://www.data-archive.ac.uk/) and via application (1946c; https://skylark.ucl.ac.uk/). The code used to run the analysis is available at https://osf.io/54unw/.

The following dataset was generated:

| Author(s) | Year | Dataset title | Dataset URL | Database and Identifier |
|---|---|---|---|---|
| Wright L | 2022 | Weakening of the cognition and height association from 1957 to 2018 | https://osf.io/54unw/ | Open Science Framework, 54unw |

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
