## [Editor Report]

This paper provides valuable evidence for a weakening of the association between cognitive ability and height from 1957 to 2018 in the UK. The authors find the strength of the association declined over this time frame. These associations were further attenuated after accounting for proxy measures of social class. This paper is a solid contribution to debates about how genetic, environmental, and social factors have affected the joint distribution of height and cognitive ability over the last 60 years.

---

## [Decision Letter]

**Decision letter after peer review:**

Thank you for submitting your article "Weakening of the cognition and height association from 1957 to 2018: findings from four British birth cohort studies" for consideration by *eLife*. Your article has been reviewed by 3 peer reviewers, and I oversaw the evaluation in my dual role of Reviewing Editor and Senior Editor. The following individual involved in review of your submission has agreed to reveal their identity: Richard Border (Reviewer #1).

As is customary in *eLife*, the reviewers have discussed their critiques with one another and with the Editors. The decision was reached by consensus. What follows below is a lightly edited compilation of the essential and ancillary points provided by reviewers in their critiques and in their interaction post-review.

Essential revisions:

The critiques by the three reviewers contain challenges or reflections related to your study's findings that the tests of cognition have behaved differently across generations and how this affects the validity of your findings. Please submit a revised version that addresses these concerns directly. Although we expect that you will address these comments in your response letter, we also need to see the corresponding revision clearly marked in the text of the manuscript. Some of the reviewers' comments may seem to be simple queries or challenges that do not prompt revisions to the text. Please keep in mind, however, that readers may have the same perspective as the reviewers. Therefore, it is essential that you amend or expand the text to clarify the narrative accordingly.

*Reviewer #1 (Recommendations for the authors):*

The authors conducted a thorough analysis of the correlation between height and measures of cognitive abilities (what are essentially IQ test components) across four cohorts of children and adolescents in the UK measured between 1957 and 2018. The authors find the strength of the association between height and cognitive measures declined over this time frame--for example, among 10- and 11-year-olds born in 1958, height explained roughly 3% of the variation in verbal reasoning scores; this dropped to approximately 0.6% among those born in 2001. These associations were further attenuated after accounting for proxy measures of social class.

The authors' analyses were performed carefully and their observations regarding declining height / cognitive measure associations are likely to be robust if we interpret their results with an important caveat: these results reflect measurements aimed at assessing cognition rather than cognition itself. The importance of this distinction is evidenced by the changing correlation structure of the cognitive measures over time. For example, age 11 verbal / math scores were correlated at >= 0.75 at the first two time points but dropped to 0.33 at the most recent time point. Similar patterns are present for the other cognitive measures and time points. The authors' conclude that such changes are unlikely to impact their primary findings, but I'm less certain. For example, one interpretation of this finding is that older cognitive measures were simply worse at indexing distinct cognitive domains and instead reflected a combination of cognitive ability together with non-specific factors relating to opportunity, health, class, etc. Further, height was historically a stronger proxy for class and economic status than it is today (e.g., by capturing adequate nutritional intake, risk for childhood disease, etc.). Together, then, previously high height / cognitive measure correlations might reflect the fact that both phenotypes previously indexed socio-economic factors to a greater extent than they might today (which is still non-negligible).

Additionally, their findings add an interesting data point to a collection of recent results suggesting that the relationship between cognitive and anthropometric measures is complex and difficult to interpret. For example, studies using genetic markers to examine shared genetic bases have virtually all relied on methods assuming mating is random, which is not the case empirically. Howe et al. (doi.org/10.1038/s41588-022-01062-7) recently reported that the ostensible genetic correlation of -.32 between years of education and BMI attenuates to -.05 when using direct-effect estimates, which should theoretically be immune to the effects of non-random mating and other confounding variables. Likewise, Keller et al. (doi.org/10.1371/journal.pgen.1003451) and Border et al. (doi.org/10.1101/2022.03.21.485215) used very different approaches to arrive at the same conclusion that ~50% of the nominal genetic correlation between IQ and height could be attributed to bivariate assortative mating rather than shared causal biological factors. Given that assortative mating on both IQ measures and height involves many other traits (not just two as assumed in such bivariate models), the true extent to which height / IQ correlations reflect causal factors is plausibly even lower than these estimates suggest. For these reasons, I do not entirely agree with the authors' review of previous findings in the introduction, where they write "recent studies have suggested that links between higher cognition and taller height can be largely explained by genetic factors", though it is certainly true that this claim has been made.

This manuscript is well-written and reflects carefully conducted research. Below are my suggestions/concerns, in no particular order.

Throughout I would suggest that the authors replace the term "cognition", with something along the lines of "cognitive measures" or "measures of cognitive ability". This might seem pedantic, but the next point highlights why I believe it is important not to conflate tools attempting to measure constructs with the constructs themselves.

Figure S4 shows a striking pattern where the inter-correlations among cognitive measures appears to sharply decline over time. E.g., age 11 verbal / math scores were correlated at >= 0.75 in the first two cohorts. This drops to 0.33 in the 2001 cohort. I would ask the authors to discuss these findings, which are currently mentioned briefly in the discussion, in greater detail. Further, I'm not sure I agree with the authors' implication that the main findings are necessarily robust to differences in tests employed over time.

For example, one interpretation of this finding is that older cognitive measures were simply worse at measuring distinct cognitive domains and instead reflected a combination of the targeted phenotype together with non-specific factors relating to opportunity, health, class, etc. To what extent to historically higher estimates simply reflect poorer measurement? Put another way, to what extent might the decrease in cognitive measure/height correlation over time reflect that IQ tests, while surely still proxies for non-cognitive measures to some extent, are better than they used to be?

Please discuss in greater detail. Further, in Figure S67, it appears that the attenuation in height/cog association after accounting for SEP is larger in the earlier cohorts than the later cohorts. If the authors have adequate power to examine this, I think it would further add to the above discussion; OTOH, if the authors think their data isn't well suited to testing this hypothesis (whether due to the fact that SEP is an imperfect measure of socio-economic status/class, or on statistical grounds), it might warrant discussion as a future direction.

As highlighted in my public comments I think it's inaccurate to state that recent studies have suggested that height/cog measure correlations can be "largely explained by genetic factors". Of the four papers cited:

(2) – certainly does make this suggestion but relies on LDSC genetic correlation, which has been found to yield extreme overestimates of rg when random mating assumptions are violated.

(3) – used an ETFD and found roughly half of phenotypic correlation attributable to bivariate assortative mating (AM). While AM is "genetic" in the fact that it reflects correlated genetic liabilities, it need not reflect any shared causal biological factors across traits, which is how I think readers usually interpret such language.

(9) – genetic correlation estimates between height and IQ were non-significant for girls, bottom of CIs were.03 and.00 for the two boys' cohorts.

(10) – These results are complex: the authors reports report that the best fitting trivariate model of height, general cognitive ability (GCA), and cortical surface area (CSA) permitted the genetic correlation between GCA and height to be constrained to zero but don't test whether the CSA / height rg can also be constrained to zero. They also report that the height / GCA association is fully mediated via CSA, but that the CSA pathway only accounts for 39% of the phenotypic correlation between height / GCA.

Together with the additional results mentioned in the public-facing review, these findings paint a complicated picture that isn't currently reflected in the introduction and discussion.

I would suggest less causal-sounding language regarding the unclear links between height and cognitive measures. E.g., in "Cognitive ability is a potentially important determinant of health" replace "determinant" with "correlate".

*Reviewer #2 (Recommendations for the authors):*

The authors use birth cohorts with extensive cognitive assessments and height measurements along with data on parental height and socioeconomic status. The authors estimate that the correlation between height and cognitive ability has approximately halved in the last 60 years.

Quantile regression results suggest that this is due to a stronger association between low cognitive ability and short stature in older cohorts, potentially due to environmental factors that cause both and that have been removed by improvements in the environment in the last 60 years.

While this is a plausible hypothesis, the evidence presented in the manuscript is unable to rule out alternative hypotheses, such as changes in assortative mating.

The results in the manuscript will be of interest to researchers investigating how genetics and environment lead to correlations between cognitive and physical/health traits, and to researchers interested in the relationship between social and health inequalities.

While my sense of the evidence presented is that there is fairly solid statistical evidence for a trend where the correlation between cognitive ability and height declines over time, there is no formal quantification of this trend nor measurement of the uncertainty in the trend.

Similarly, the quantile regression plots in Figure 2 appear to show a trend across the height deciles for the two oldest cohorts, but no quantification of how strong this is nor what uncertainty exists is calculated. Furthermore, if the apparent trend in the quantile regression plots is true, wouldn't this imply a non-linear association between height and cognitive ability for the older cohorts? Can this be seen in the scatterplots or in a non-linear regression?

I think the authors could have done more with their data to investigate the contribution of assortative mating to the observed trend. Looking at Figure S4, it looks like the correlation between mother's education and father's height in the 2001 cohort is substantially lower than for previous cohorts. While cognitive ability may not be available for parents, one could look at, for example, father's education and mother's height across the cohorts and see if there is a downward trend in correlation.

---

## [Author Response]

Reviewer #1 (Recommendations for the authors):The authors conducted a thorough analysis of the correlation between height and measures of cognitive abilities (what are essentially IQ test components) across four cohorts of children and adolescents in the UK measured between 1957 and 2018. The authors find the strength of the association between height and cognitive measures declined over this time frame--for example, among 10- and 11-year-olds born in 1958, height explained roughly 3% of the variation in verbal reasoning scores; this dropped to approximately 0.6% among those born in 2001. These associations were further attenuated after accounting for proxy measures of social class.The authors' analyses were performed carefully and their observations regarding declining height / cognitive measure associations are likely to be robust if we interpret their results with an important caveat: these results reflect measurements aimed at assessing cognition rather than cognition itself. The importance of this distinction is evidenced by the changing correlation structure of the cognitive measures over time. For example, age 11 verbal / math scores were correlated at >= 0.75 at the first two time points but dropped to 0.33 at the most recent time point. Similar patterns are present for the other cognitive measures and time points. The authors' conclude that such changes are unlikely to impact their primary findings, but I'm less certain. For example, one interpretation of this finding is that older cognitive measures were simply worse at indexing distinct cognitive domains and instead reflected a combination of cognitive ability together with non-specific factors relating to opportunity, health, class, etc. Further, height was historically a stronger proxy for class and economic status than it is today (e.g., by capturing adequate nutritional intake, risk for childhood disease, etc.). Together, then, previously high height / cognitive measure correlations might reflect the fact that both phenotypes previously indexed socio-economic factors to a greater extent than they might today (which is still non-negligible).

We agree, it is possible that our results could in principle be explained by changes to the measures. We have provided further analysis to attempt to inform the likelihood of this suggestion and have expanded our discussion of this issue (Discussion, explanation of findings section; copied below).

First, we conducted additional sensitivity analysis repeating our main analysis using cognition measures in which the number of response options was set to be the same for each test (the lowest common denominator across all cohorts). This was tested in two separate approaches: 1) by reducing the number of categories to the same number in each cohort; and 2) or by picking a random sample of question items for each category. Our main findings were unchanged: described in “Additional and sensitivity analyses” section, FiguresS20-S21.

Regarding the suggestion that “high height / cognitive measure correlations might reflect the fact that both phenotypes previously indexed socio-economic factors to a greater extent than they might today” – we sought to account for this by adjustment for measured indicators of socioeconomic position, and found the trend remained after adjustment. As in other observational studies we cannot fully rule out the possibility of residual confounding however (Discussion, Explanation of findings paragraph 2).

“The multi-purpose and multidisciplinary cohorts used cognition tests which differed slightly in each cohort. It is therefore possible that differences in testing could have either: 1) entirely generated the pattern of results we observed, such that if identical tests were used the association between cognition and height would otherwise have been identical in each cohort; in contrast to previous findings which reported using identical tests^20^; or 2) biased our results, such that if identical tests were used the decline in association between cognition and height would have been less marked than we reported. While we cannot directly falsify this alternative hypothesis given our reliance on historical data sources, a number of lines of reasoning suggest that the first scenario is unlikely. First, our results were similar when using 4 different cognitive tests (spanning mathematical and verbal reasoning); any bias which generated the results we observed should be similarly present across all 4 tests. Other things being equal, one would expect that more discriminatory tests (i.e., those with a greater number of responses) would have higher accuracy and thus better index cognition. Our results were similar when the youngest cohort had similar numbers of unique scores in cognitive tests compared with the oldest cohort (Verbal @ 11 years: n=41 in 1946c, n=40 in 2001c) and fewer unique scores (Maths @ 7/11: n=51 in 1946c, n=21 in 2001c). Our results were also similar in sensitivity analyses in which the number of response options were set to be the same in each cohort. Higher random measurement error in the independent variable (cognition) would lead to weakened observed associations with the outcome (height),^52^ yet we do not a-priori anticipate that this such error was higher in younger across all tests in such a manner that would have led to the correlation we observed.

Ensuring comparability of exposure is a major challenge across such large timespans. Reassuringly, our results are consistent with those from a previous study which reported consistent tests being used (from 1939-1967).^20^ However, even seemingly identical require modification across time (e.g., for verbal reasoning/vocabulary there is typically a need to adapt question items due to societal and cultural changes over time in vocabulary and numerical use); further, changes to education such as increases in testing may have led to increasing preparedness and familiarity with testing than in the past even where identical tests are used.

Interestingly, we observed a marked reduction in the correlation between cognitive tests across time (e.g., between verbal and maths scores). This trend has been reported in previous studies^53 54^ and warrants future investigation; it is consistent with evidence that IQ gains across time seemingly differ by cognitive domain,^45^ potentially capturing differences across time in cognitive skill use and development in the population. Previous studies using three (1958-2001c) of the included cohorts have also reported changing associations between cognition (verbal test scores at 10/11 years) and other traits: a declining negative association with birth weight^19^ and a change in direction of association with maternal age (from negative to positive);^55^ each finding has plausible explanations based on changes across time in relevant societal phenomena (improved medical conditions^19^ and changes in parental characteristics,^55^ respectfully), yet also cannot conclusively falsify the notion that differences in tests used influences the results obtained. In this paper, we used multiple tests and sensitivity analyses to attempt to address this.”

Additionally, their findings add an interesting data point to a collection of recent results suggesting that the relationship between cognitive and anthropometric measures is complex and difficult to interpret. For example, studies using genetic markers to examine shared genetic bases have virtually all relied on methods assuming mating is random, which is not the case empirically. Howe et al. (doi.org/10.1038/s41588-022-01062-7) recently reported that the ostensible genetic correlation of -.32 between years of education and BMI attenuates to -.05 when using direct-effect estimates, which should theoretically be immune to the effects of non-random mating and other confounding variables. Likewise, Keller et al. (doi.org/10.1371/journal.pgen.1003451) and Border et al. (doi.org/10.1101/2022.03.21.485215) used very different approaches to arrive at the same conclusion that ~50% of the nominal genetic correlation between IQ and height could be attributed to bivariate assortative mating rather than shared causal biological factors. Given that assortative mating on both IQ measures and height involves many other traits (not just two as assumed in such bivariate models), the true extent to which height / IQ correlations reflect causal factors is plausibly even lower than these estimates suggest. For these reasons, I do not entirely agree with the authors' review of previous findings in the introduction, where they write "recent studies have suggested that links between higher cognition and taller height can be largely explained by genetic factors", though it is certainly true that this claim has been made.

We have revised our introduction to better reflect the complexity of previous findings and to note that this claim. Please also see our response below.

This manuscript is well-written and reflects carefully conducted research. Below are my suggestions/concerns, in no particular order.Throughout I would suggest that the authors replace the term "cognition", with something along the lines of "cognitive measures" or "measures of cognitive ability". This might seem pedantic, but the next point highlights why I believe it is important not to conflate tools attempting to measure constructs with the constructs themselves.

We have checked the text throughout and additionally used the term ‘cognition scores’ to try to address this point in several instances. In other instances, we felt that the word cognition was a succinct summary of the intended meaning (ie, a necessary tool used to capture the otherwise unmeasured construct); if the editorial team felt that we should make further amendments we would be happy to consider this.

Figure S4 shows a striking pattern where the inter-correlations among cognitive measures appears to sharply decline over time. E.g., age 11 verbal / math scores were correlated at >= 0.75 in the first two cohorts. This drops to 0.33 in the 2001 cohort. I would ask the authors to discuss these findings, which are currently mentioned briefly in the discussion, in greater detail. Further, I'm not sure I agree with the authors' implication that the main findings are necessarily robust to differences in tests employed over time.

We have added to our discussion of this point. We are working on a separate paper which tackles this correlation in greater detail – it is consistent with studies in the Flynn effect literature, which suggest that increases across time have been divergent across cognitive domain (thus implying that correlations would weaken across time). Please see above regarding the general issue of the robustness of our findings to differences in tests used.

For example, one interpretation of this finding is that older cognitive measures were simply worse at measuring distinct cognitive domains and instead reflected a combination of the targeted phenotype together with non-specific factors relating to opportunity, health, class, etc. To what extent to historically higher estimates simply reflect poorer measurement? Put another way, to what extent might the decrease in cognitive measure/height correlation over time reflect that IQ tests, while surely still proxies for non-cognitive measures to some extent, are better than they used to be?

Please see above and the revisions for a broader discussion of this possibility. On balance we think it is unlikely for this to be the sole explanation for the pattern of results that we find across 4 separate tests (and sensitivity analyses).

Please discuss in greater detail. Further, in Figure S67, it appears that the attenuation in height/cog association after accounting for SEP is larger in the earlier cohorts than the later cohorts. If the authors have adequate power to examine this, I think it would further add to the above discussion; OTOH, if the authors think their data isn't well suited to testing this hypothesis (whether due to the fact that SEP is an imperfect measure of socio-economic status/class, or on statistical grounds), it might warrant discussion as a future direction.

We have added an additional supplementary information (Figure 3) which provides numerical figures on the degree of attenuation and discuss this (Discussion, Explanation of Findings paragraph 2).

As highlighted in my public comments I think it's inaccurate to state that recent studies have suggested that height/cog measure correlations can be "largely explained by genetic factors".

We have updated the introduction to better reflect findings from previous work (see Introduction paragraph 2). Please see below/the revised introduction (paragraph 2).

“Recent studies have suggested that links between higher cognition and taller height can be largely or to a sizable extent explained by shared genetic influences,2 3 9 10 with others suggesting a role of assortative mating.3 11 Keller et al. (2013) for example estimated that approximately half of the genetic association found in a single cohort was accounted for by assortative mating, and half by shared genetic factors.3”

Below we have provided quotes from each paper in support of their citation to support the statements made. We have refrained from additional discussion of methodological detail to retain the focus of the paper and ensure the introduction remained on-topic (eg, we do not test measures of brain structure (ref 10)).

Of the four papers cited:(2) – certainly does make this suggestion but relies on LDSC genetic correlation, which has been found to yield extreme overestimates of rg when random mating assumptions are violated.

Marioni RE, Batty GD, Hayward C, et al. Common genetic variants explain the majority of the correlation between height and intelligence: the generation Scotland study. Behav Genet 2014;44(2):91-96.

Statement suggesting explained by genetic factors:

“This study identified a modest genetic correlation between height and intelligence with the majority of the phenotypic correlation being explained by shared genetic influences.”

(3) – used an ETFD and found roughly half of phenotypic correlation attributable to bivariate assortative mating (AM). While AM is "genetic" in the fact that it reflects correlated genetic liabilities, it need not reflect any shared causal biological factors across traits, which is how I think readers usually interpret such language.

Keller MC, Garver-Apgar CE, Wright MJ, et al. The genetic correlation between height and IQ: Shared genes or assortative mating? PLoS genetics 2013;9(4):e1003451.

Statement suggesting explained by genetic factors + assortative mating:

“We used this model to demonstrate that the phenotypic correlation between two potentially sexually selected traits in humans, IQ and height, is largely genetic in nature, and that both shared genes and assortative mating contribute importantly to it”

(9) – genetic correlation estimates between height and IQ were non-significant for girls, bottom of CIs were.03 and.00 for the two boys' cohorts.

Silventoinen K, Iacono WG, Krueger R, et al. Genetic and environmental contributions to the association between anthropometric measures and IQ: a study of Minnesota twins at age 11 and 17. Behav Genet 2012;42(3):393-401.

Statement suggesting explained by genetic factors:

“head circumference and total height showed stronger associations with IQ than anthropometric measures focused on specific body parts. We found some evidence that these associations were mainly genetic origin, which may indicate the role of endocrinological factors or genetically based susceptibility to environmental stressors shared by family members.”

(10) – These results are complex: the authors reports report that the best fitting trivariate model of height, general cognitive ability (GCA), and cortical surface area (CSA) permitted the genetic correlation between GCA and height to be constrained to zero but don't test whether the CSA / height rg can also be constrained to zero. They also report that the height / GCA association is fully mediated via CSA, but that the CSA pathway only accounts for 39% of the phenotypic correlation between height / GCA.Together with the additional results mentioned in the public-facing review, these findings paint a complicated picture that isn't currently reflected in the introduction and discussion.

Vuoksimaa E, Panizzon MS, Franz CE, et al. Brain structure mediates the association between height and cognitive ability. Brain Structure and Function 2018;223(7):3487-94.

Statement suggesting explained by genetic factors:

“Both height and GCA have substantial heritability and their association is largely due to shared genetic effects”

I would suggest less causal-sounding language regarding the unclear links between height and cognitive measures. E.g., in "Cognitive ability is a potentially important determinant of health" replace "determinant" with "correlate".

We agree it is absolutely essential to be very careful with our use of causal language. We believe that “potentially important determinant” reflects the implied causal link and the uncertainty. We refer to https://ajph.aphapublications.org/doi/10.2105/AJPH.2018.304337 for an illustration of the benefits of being explicit regarding causal links, even if this is challenging using observational data.

We have checked the manuscript throughout regarding the causal language used and edited where it was felt that our intended meaning was unclear. On reflection we have opted to retain the causal language where causal links are intended to be implied. We would be happy to revise further if we have overstated or misled in our claims re causation.

Reviewer #2 (Recommendations for the authors):The authors use birth cohorts with extensive cognitive assessments and height measurements along with data on parental height and socioeconomic status. The authors estimate that the correlation between height and cognitive ability has approximately halved in the last 60 years.Quantile regression results suggest that this is due to a stronger association between low cognitive ability and short stature in older cohorts, potentially due to environmental factors that cause both and that have been removed by improvements in the environment in the last 60 years.While this is a plausible hypothesis, the evidence presented in the manuscript is unable to rule out alternative hypotheses, such as changes in assortative mating.The results in the manuscript will be of interest to researchers investigating how genetics and environment lead to correlations between cognitive and physical/health traits, and to researchers interested in the relationship between social and health inequalities.While my sense of the evidence presented is that there is fairly solid statistical evidence for a trend where the correlation between cognitive ability and height declines over time, there is no formal quantification of this trend nor measurement of the uncertainty in the trend.

We now include additional statistical tests to compare estimates in each cohort (Figure 3). We have opted to include this in supplemental material given the large number of tests included already.

Similarly, the quantile regression plots in Figure 2 appear to show a trend across the height deciles for the two oldest cohorts, but no quantification of how strong this is nor what uncertainty exists is calculated. Furthermore, if the apparent trend in the quantile regression plots is true, wouldn't this imply a non-linear association between height and cognitive ability for the older cohorts? Can this be seen in the scatterplots or in a non-linear regression?

We included 95% confidence intervals in our quantile regression analyses which provide an indication of uncertainty. We believe that given the substantial amount of analyses (across 4 historical cohorts and 4 cognition tests; 23 supplemental results) further work would be best placed to undertake additional statistical exploration of both quantile regression and non-linear associations. We would be happy to reconsider this if requested.

I think the authors could have done more with their data to investigate the contribution of assortative mating to the observed trend. Looking at Figure S4, it looks like the correlation between mother's education and father's height in the 2001 cohort is substantially lower than for previous cohorts. While cognitive ability may not be available for parents, one could look at, for example, father's education and mother's height across the cohorts and see if there is a downward trend in correlation.

We now include in Figure 2 cross-cohort investigation of the correlation between parental height and maternal education. We find that the correlation is similar across 1946c, 1958c, and 1970c, yet is weaker in 2001c. We comment on this in the paper (see revised discussion, explanation of findings section). Interpretation of these results is complicated by measurement error in parental education (typically reported for both parents by mothers). Further, interpretation may be further complicated by reductions in the socioeconomic patterning of height across time (see https://www.thelancet.com/journals/lanpub/article/PIIS2468-2667(18)30045-8/fulltext). Future would which focuses on assortative mating could investigate these issues.